# How to Select Which Active Learning Strategy is Best Suited for Your Specific Problem and Budget

**Guy Hacohen**[†‡]**, Daphna Weinshall**[†]
School of Computer Science & Engineering[†]
Edmond and Lily Safra Center for Brain Sciences[‡]
The Hebrew University of Jerusalem
Jerusalem 91904, Israel
{guy.hacohen,daphna}@mail.huji.ac.il

## Abstract

In the domain of Active Learning (AL), a learner actively selects which unlabeled examples to seek labels from an oracle, while operating within predefined budget constraints. Importantly, it has been recently shown that distinct query strategies are better suited for different conditions and budgetary constraints. In practice, the determination of the most appropriate AL strategy for a given situation remains an open problem. To tackle this challenge, we propose a practical derivative-based method that dynamically identifies the best strategy for a given budget. Intuitive motivation for our approach is provided by the theoretical analysis of a simplified scenario. We then introduce a method to dynamically select an AL strategy, which takes into account the unique characteristics of the problem and the available budget. Empirical results showcase the effectiveness of our approach across diverse budgets and computer vision tasks.

## 1 Introduction

Active learning emerged as a powerful approach for promoting more efficient and effective learning outcomes. In the traditional supervised learning framework, active learning enables the learner to actively engage in the construction of the labeled training set by selecting a fixed-sized subset of unlabeled examples for labeling by an oracle, where the number of labels requested is referred to as the *budget*. Our study addresses the task of identifying in advance the most appropriate active learning strategy for a given problem and budget.

The selection of an active learning strategy is contingent on both the learner's inductive biases and the nature of the problem at hand. But even when all this is fixed, recent research has shown that the most suited active learning strategy varies depending on the size of the budget. When the budget is large, methods based on uncertainty sampling are most effective. When the budget is small, methods based on typicality are most suitable (see Fig. 1). In practice, determining the appropriate active learning strategy based on the budget size is challenging, as a specific budget can be considered either small or large depending on the problem at hand. This challenge is addressed in this paper.

Specifically, we start by analyzing a simplified theoretical framework (Section 2), for which we can explicitly select the appropriate AL strategy using a derivative-based test. Motivated by the analysis of this model, we propose SelectAL (Section 3), which incorporates a similar derivative-based test to select between active learning strategies. SelectAL aims to provide a versatile solution for any budget, by identifying the budget domain of the problem at hand and picking an appropriate AL method from the set of available methods. SelectAL is validated through an extensive empirical study using several vision datasets (Section 4). Our results demonstrate that SelectAL is effective in identifying the best active learning strategy for any budget, achieving superior performance across all budget ranges.

37th Conference on Neural Information Processing Systems (NeurIPS 2023).

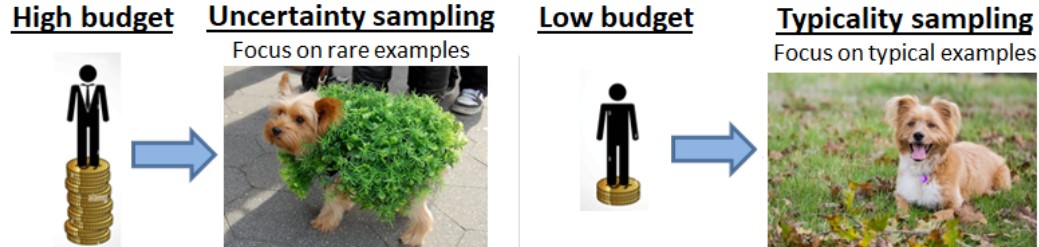

Figure 1: Qualitatively different AL strategies are suited for different budgets. SelectAL determines in advance which family of strategies should be used for the problem at hand. Left: when the labeled training set (aka budget) is large, uncertainty sampling, which focuses on unusual examples, provides the most added value. Right: when the budget is small, the learner benefits most from seeing characteristic examples.

**Relation to prior work.** Active learning has been an active area of research in recent years [32, 34]. The traditional approach to active learning, which is prevalent in deep learning and recent work, focuses on identifying data that will provide the greatest added value to the learner based on what it already knows. This is typically achieved through the use of uncertainty sampling [11, 24, 31, 36], diversity sampling [8, 20, 33, 35], or a combination of both [1, 10, 22].

However, in small-budget settings, where the learner has limited prior knowledge and effective training is not possible before the selection of queries, such active learning methods fail [4, 27]. Rather, [13, 30, 37] have shown that in this domain, a qualitatively different family of active learning methods should be used. These methods are designed to be effective in this domain and tend to seek examples that can be easily learned rather than confusing examples [5, 25, 39].

With the emergence of this distinction between two separate families of methods to approach active learning, the question arises as to which approach should be preferred in a given context, and whether it is possible to identify in advance which approach would be most effective. Previous research has explored methods for selecting between various AL strategies [2, 19, 29, 40]. However, these studies assumed knowledge of the budget, which is often unrealistic in practice. To our knowledge, SelectALis the first work that addresses this challenge.

We note that several recent works focused on the distinctions between learning with limited versus ample data, but within the context of non-active learning [6, 9, 12, 14, 15, 16, 21, 26]. These studies often highlight qualitative differences in model behavior under varying data conditions. While our results are consistent with these observations, our focus here is to suggest a practical approach to identify the budget regime in advance, allowing for the selection of the most appropriate AL strategy.

## 2 Theoretical analysis

The aim of this section is to derive a theoretical decision rule that can be translated into a practical algorithm, thus enabling practitioners to make data-driven decisions on how to allocate their budget in active learning scenarios. Given an AL scenario, this rule will select between a high-budget approach, a low-budget approach, or a blend of both. In Section 3, the theoretical result motivates the choice of a decision rule, in a practical method that selects between different active learning strategies.

To develop a test that can decide between high and low-budget approaches for active learning, we seek a theoretically sound framework in which both approaches can be distinctly defined, and show how they can be beneficial for different budget ranges. To this end, we adopt the theoretical framework introduced by [13]. While this framework is rather simplistic, it allows for precise formulation of the distinctions between the high and low-budget approaches. This makes it possible to derive a precise decision rule for the theoretical case, giving some insights for the practical case later on.

In Section 2.1, we establish the necessary notations and provide a brief summary of the theoretical framework used in the analysis, emphasizing the key assumptions and highlighting the results that are germane to our analysis. We then establish in Section 2.2 a derivative-based test, which is a novel contribution of our work. In Section 2.3, we utilize this test to derive an optimal active learning strategy for the theoretical framework. To conclude, in Section 2.4 we present an empirical validation of these results, accompanied by visualizations for further clarification.

## 2.1 Preliminaries

**Notations.** We consider an active learning scenario where a learner, denoted by $\mathcal{L}$, is given access to a set of labeled examples $\mathbb{L}$, and a much larger pool of unlabeled examples $\mathbb{U}$. The learner's task is to select a fixed-size subset of the unlabeled examples, referred to as the *active set* and denoted by $\mathbb{A} \subseteq \mathbb{U}$, and obtain their labels from an oracle. These labeled examples are then used for further training of the learner. The goal is to optimally choose the active set $\mathbb{A}$, such that the overall performance of learner $\mathcal{L}$ trained on the combined labeled dataset $\mathbb{T} = \mathbb{A} \cup \mathbb{L}$ is maximal.

We refer to the total number of labeled examples as the *budget*, denoted by $B = |\mathbb{T}| = |\mathbb{A} \cup \mathbb{L}|$. This concept of budget imposes a critical constraint on the active learning process, as the learner must make strategic selections within the limitations of the budget.

**Model definition and assumptions.** We now summarize the abstract theoretical framework established by [13], which is used in our analysis. While this framework contains simplistic assumptions, its insights are shown empirically to be beneficial in real settings. This framework considers two independent general learners $\mathcal{L}_{low}$ and $\mathcal{L}_{high}$, each trained on a different data distribution $\mathcal{D}_{low}$ and $\mathcal{D}_{high}$ respectively. Intuitively, one can think of $\mathcal{D}_{low}$ as a distribution that is easier to learn than $\mathcal{D}_{high}$, such that $\mathcal{L}_{low}$ requires fewer examples than $\mathcal{L}_{high}$ to achieve a similar generalization error.

The number of training examples each learner sees may vary between the learners. To simplify the analysis, the framework assumes that data naturally arrives from a mixture distribution $\mathcal{D}$, in which an example is sampled from $\mathcal{D}_{low}$ with probability $p$ and from $\mathcal{D}_{high}$ with probability $1 - p$.

We examine the mean generalization error of each learner denoted by $E_{low}, E_{high} : \mathbb{R} \to [0, 1]$ respectively, as a function of the number of training examples it sees. The framework makes several assumptions about the form of these error functions: (i) Universality: Both $E_{low}$ and $E_{high}$ take on the same universal form $E(x)$, up to some constant $\alpha > 0$, where $E(x) = E_{low}(x) = E_{high}(\alpha x)$. (ii) Efficiency: $E(x)$ is continuous and strictly monotonically decreasing, namely, on average each learner benefits from additional examples. (iii) Realizability: $\lim_{x \to \infty} E(x) = 0$, namely, given enough examples, the learners can perfectly learn the data.

To capture the inherent choice in active learning, a family of general learners is considered, each defined by a linear combination of $\mathcal{L}_{low}$ and $\mathcal{L}_{high}$. [13] showed that when fixing the number of examples available to the mixture model, i.e, fixing the budget $B$, it is preferable to sample the training data from a distribution that differs from $\mathcal{D}$. Specifically, it is preferable to skew the distribution towards $\mathcal{D}_{low}$ when the budget is low and towards $\mathcal{D}_{high}$ when the budget is high.

## 2.2 Derivative-based query selection decision rule

We begin by asking whether it is advantageous to skew the training distribution towards the low or high-budget distribution. Our formal analysis gives a positive answer. It further delivers a derivative-based test, which can be computed in closed form for the settings above. Importantly, our empirical results (Section 4) demonstrate the effectiveness of an approximation of this test in deep-learning scenarios, where the closed-form solution could not be obtained.

We begin by defining two pure query selection strategies – one that queries examples only from $\mathcal{D}_{low}$, suitable for the low-budget regime, and a second that queries examples only from $\mathcal{D}_{high}$, suitable for the high-budget regime. Given a fixed budget $B$, we analyze the family of strategies obtained by a linear combination of these pure strategies, parameterized by $q \in [0, 1]$, in which $qB$ points are sampled from $\mathcal{D}_{low}$ and $(1 - q)B$ points are sampled using $\mathcal{D}_{high}$.

The mean generalization error of combined learner $\mathcal{L}$ on the original distribution $\mathcal{D}$, where $\mathcal{L}$ is trained on $B$ examples picked by a mixed strategy $q$, is denoted $E_{\mathcal{L}}(B, q) = p \cdot E(qB) + (1-p) \cdot E(\alpha(1-q)B)$, as the combined learner simply sends $qB$ examples to $\mathcal{L}_{low}$ and $(1 - q)B$ examples to $\mathcal{L}_{high}$. By differentiating this result (see derivation in Suppl. A), we can find an optimal strategy for this family, and denote it by $\hat{q}$. This strategy is defined as the one that delivers the lowest generalization error when using a labeled set of size $B$. The strategy is characterized by $\hat{q} \in [0, 1]$, implicitly defined as follows:

$$\frac{E^{'}\left(\hat{q}B\right)}{E^{'}\left(\alpha\left(1 - \hat{q}\right)B\right)} = \frac{\alpha\left(1 - p\right)}{p}. \tag{1}$$

If Eq. (1) has a unique and minimal solution for $\hat{q}$, it defines an optimal mixed strategy $\hat{q}_E(B, p, \alpha)$ for a given set of problem parameters $\{B, p, \alpha\}$. As $p$ and $\alpha$ are fixed for any specific problem, we get that different budgets $B$ may require different optimal AL strategies. Notably, if budget $B$ satisfies $\hat{q}_E(B, p, \alpha) = p$, then the optimal strategy for it is equivalent[1] to selecting samples directly from the original distribution $\mathcal{D}$. We refer to such budgets as $B_{eq}$. With these budgets, active learning is no longer helpful. While $B_{eq}$ may not be unique, for the sake of simplicity, we assume that it is unique in our analysis, noting that the general case is similar but more cumbersome.

We propose to use Eq. (1) as a decision rule, to determine whether a low-budget or high-budget strategy is more suitable for the current budget $B$. Specifically, given budget $B$, we can compute $\hat{q}_E(B, p, \alpha)$ and $B_{eq}$ in advance and determine which AL strategy to use by comparing $B$ and $B_{eq}$. In the current framework, this rule is guaranteed to deliver an optimal linear combination of strategies, as we demonstrate in Section 2.3. Visualization of the model's error as a function of the budget, for different values of $q$, can be seen in Fig. 2a.

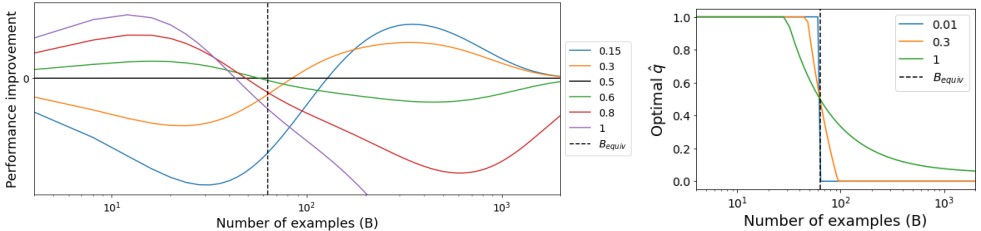

(a) Accuracy gain of mixed strategies $q$, where mixture coefficient $q$ is indicated in the legend.

(b) Optimal $\hat{q}_a(B, p, \alpha)$.

Figure 2: Visualization of $E_{\mathcal{L}}(B, q)$, where training set of size $B$ is selected for different values of $q$. We adopt the example from [13], choosing $E(x) = e^{-ax}$, $a = 0.1$, $p = \frac{1}{2}$ and $\alpha = 0.05$, as exponential functions approximate well the error functions of real networks. (a) A plot of accuracy gain when using strategy $q$ (indicated in legend) as compared to random query selection: $E(p) - E(q)$, as a function of budget $B$. Since $p = \frac{1}{2}$, the plot corresponding to $q = \frac{1}{2}$ is always 0. (b) Plots of $\hat{q}$ as a function of $B$. $B_{eq}$, which corresponds to $\hat{q} = \frac{1}{2}$, is indicated by a vertical dashed line. Each plot corresponds to a different fraction $\frac{|\mathbb{A}|}{B}$ (see legend).

## 2.3  Best mixture of active learning strategies

With the aim of active learning (AL) in mind, our objective is to strategically select the active set $\mathbb{A}$ in order to maximize the performance of learner $\mathcal{L}$ when trained on all available labels $\mathbb{T} = \mathbb{A} \cup \mathbb{L}$. To simplify the analysis, we focus on the initial AL round, assuming that $\mathbb{L}$ is sampled from the underlying data distribution $\mathcal{D}$. The analysis of subsequent rounds can be done in a similar manner.

To begin with, let us consider the entire available label set $\mathbb{T} = \mathbb{A} \cup \mathbb{L}$. Since $\mathbb{T}$ comprises examples from $\mathcal{D}$, $\mathcal{D}_{low}$ and $\mathcal{D}_{high}$, we can represent it using the (non-unique) notation $\mathcal{S}(r_{rand}, r_{low}, r_{high})$, where $r_{rand}, r_{low}, r_{high}$ denote the fractions of $\mathbb{T}$ sampled from each respective distribution. Based on the definitions of $q$ and $\hat{q}$, we can identify an optimal combination for $\mathbb{T}$, denoted $\mathbb{T}^*$:

$$\mathbb{T}^* = \begin{cases} \mathcal{S}(1 - \hat{r}, \hat{r}, 0) & \hat{q} > p \implies B < B_{eq} \\ \mathcal{S}(1, 0, 0) & \hat{q} = p \implies B = B_{eq} \\ \mathcal{S}(1 - \hat{r}, 0, \hat{r}) & \hat{q} < p \implies B > B_{eq} \end{cases} \qquad \hat{r} = \begin{cases} \frac{1}{1-p}(\hat{q} - p) & \hat{q} > p \\ \frac{1}{1-p}(p - \hat{q}) & \hat{q} < p \end{cases} \quad (2)$$

**Attainable optimal mixed strategy.** It is important to note that $\mathbb{T} = \mathbb{A} \cup \mathbb{L}$, where the active learning (AL) strategy can only influence the sampling distribution of $\mathbb{A}$. Consequently, not every combination of $\mathbb{T}$ is attainable. In fact, the feasibility of the optimal combination relies on the size of the set $\mathbb{A}$. Specifically, utilizing (1), we can identify two thresholds, $B_{low}$ and $B_{high}$, such that if $B < B_{low} \leq B_{eq}$ or $B > B_{high} \geq B_{eq}$, it is not possible to achieve the optimal $\mathbb{T}^* = \mathbb{A}^* \cup \mathbb{L}$. The derivation of both thresholds can be found in Suppl. A.

---

[1]Identity (in probability) is achieved when $B \to \infty$.

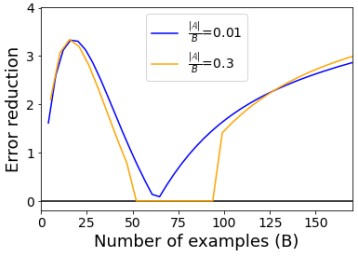

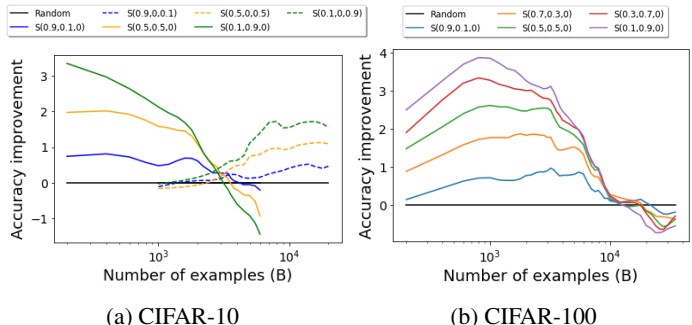

Figure 3: Visualization of strategy (3) using the example from Fig. 2. We plot the gain in error reduction as compared to random query selection. We show results with active sets that are 30% and 1% of $B$. The smaller the active set is, the closer the performance is to the pure optimal strategy.

(a) CIFAR-10      (b) CIFAR-100

Figure 4: Empirical validation of the theoretical results. We plot the accuracy gains of 20 ResNet-18 networks trained using strategy (3), compared to using no active learning at all. We used TypiClust as $S_{low}$ and BADGE as $S_{high}$. We see that as predicted by the theoretical analysis, the best mixture coefficient $r$ increases as the difference $|B - B_{eq}|$ increases.

With (2) and the thresholds above, we may conclude that the optimal combination of $\mathbb{A}^*$ is:

$$
\mathbb{A}^* = \begin{cases} S(0,1,0) & B < B_{low} \\ S(1 - \hat{r}\frac{|\mathbb{T}|}{|\mathbb{A}|}, \hat{r}\frac{|\mathbb{T}|}{|\mathbb{A}|}, 0) & B_{low} \leq B < B_{eq} \\ S(1,0,0) & B = B_{eq} \\ S(1 - \hat{r}\frac{|\mathbb{T}|}{|\mathbb{A}|}, 0, \hat{r}\frac{|\mathbb{T}|}{|\mathbb{A}|}) & B_{high} \geq B > B_{eq} \\ S(0,0,1) & B > B_{high} \end{cases} \quad (3) \quad \xrightarrow{|\mathbb{A}| \to 0} \mathbb{A}^* = \begin{cases} S(0,1,0) & B < B_{low} \\ S(1,0,0) & B = B_{eq} \\ S(0,0,1) & B > B_{high} \end{cases} \quad (4)
$$

In other words, we observe that the optimal strategy (2) is only attainable in non-extreme scenarios. Specifically, in cases of very low budgets ($B < B_{low}$), it is optimal to sample the active set purely from $\mathcal{D}_{low}$ because more than $|\mathbb{A}|$ points from $\mathcal{D}_{low}$ are needed to achieve optimal performance. Similarly, in situations of very high budgets ($B > B_{high}$), it is optimal to select the active set solely from $\mathcal{D}_{high}$ since more than $|\mathbb{A}|$ points from $\mathcal{D}_{high}$ are necessary for optimal performance. We note that even if $B_{eq}$ is not unique, in the limit of $|\mathbb{A}| \to 0$, (3)-(4) are still locally optimal, segment by segment.

**Small active set $\mathbb{A}$.** Upon examining the definitions of $B_{low}$ and $B_{high}$ (refer to Suppl. A), we observe that $\lim_{|\mathbb{A}| \to 0} B_{low} = \lim_{|\mathbb{A}| \to 0} B_{high} = B_{eq}$. Consequently, as indicated in (4), the optimal mixed strategy, in this scenario, actually becomes a pure strategy. When the budget is low, the entire active set $\mathbb{A}$ should be sampled from $\mathcal{D}_{low}$. Similarly, when the budget is high, $\mathbb{A}$ should be sampled solely from $\mathcal{D}_{high}$. In the single point where $B_{low} = B_{high} = B_{eq}$, $\mathbb{A}$ should be sampled from $\mathcal{D}$. Fig. 2b provides a visualization of these strategies as the size of $\mathbb{A}$ decreases.

**Iterative querying.** When $|\mathbb{A}|$ is not small enough to justify the use of the limiting pure strategy (4), we propose to implement query selection incrementally. By repeatedly applying (4) to smaller segments of length $m$, we can iteratively construct $\mathbb{A}$, with each iteration becoming more computationally feasible. This strategy not only enhances robustness but also yields comparable performance to the mixed optimal strategy (3), as demonstrated in Section 2.4.

Based on the analysis presented above, it becomes apparent that in practical scenarios, **the optimal combination of active learning strategies can be achieved by sequentially sampling from pure strategies**, while utilizing a derivative-based test to determine the most effective strategy at each step. This concept forms the core motivation behind the development of the practical algorithm SelectAL, which is presented in Section 3.

## 2.4 Validation and visualization of theoretical results

**Visualization.** In Fig. 3, we illustrate the error of the strategies defined in (3) using the same exponential example as depicted in Fig. 2. The orange curve represents a relatively large active set, where $|\mathbb{A}|$ is equal to 30% of the budget $B$, while the blue curve represents a significantly smaller active set, where $|\mathbb{A}|$ is equal to 1% of the budget $B$. It is evident that as the size of $\mathbb{A}$ decreases, the discrepancy between the optimal strategy and the attainable strategy diminishes.

Fig. 2b shows the optimal mixture coefficient $\hat{q}$ as a function of the budget size $B$ for various values of $|\mathbb{A}|$, namely $1\%$, $30\%$, and $100\%$ of $B$. According to our analysis, as the size of $|\mathbb{A}|$ decreases, the optimal $\hat{q}$ should exhibit a more pronounced step-like behavior. This observation suggests that in the majority of cases, it is possible to sample the entire active set $\mathbb{A}$ from a single strategy rather than using a mixture of strategies.

**Validation.** Our theoretical analysis uses a mixture model of idealized general learners. We now validate that similar phenomena occur in practice when training deep networks on different computer vision tasks. Since sampling from $\mathcal{D}_{low}$ and $\mathcal{D}_{high}$ is not feasible in this case, we instead choose low and high-budget deep AL strategies from the literature. Specifically, we choose *TypiClust* as the low-budget strategy and *BADGE* as the high-budget strategy, as explained in detail in Section 4.1. We note that other choices for the low and high-budget strategies yield similar qualitative results, as is evident from Tables 1-2. Fig. 4 shows results when training 20 ResNet-18 networks using mixed strategies $S(1 - r\frac{|\mathbb{T}|}{|\mathbb{A}|}, 0, r\frac{|\mathbb{T}|}{|\mathbb{A}|})$ and $S(1 - r\frac{|\mathbb{T}|}{|\mathbb{A}|}, r\frac{|\mathbb{T}|}{|\mathbb{A}|}, 0)$ on CIFAR-10 and CIFAR-100. We compare the mean performance of each strategy to the performance of 20 ResNet-18 networks trained with the random query selection strategy $S(1, 0, 0)$.

Inspecting these results, we observe similar trends to those shown in the theoretical analysis. As the budget increases, the most beneficial value of mixture coefficient $r$ decreases until a certain transition point (corresponding to $B_{eq}$). From this transition point onward, the bigger the budget is, the more beneficial it is to select additional examples from one of the high-budget strategies. As in the theoretical analysis, when the budget is low it is beneficial to use a pure low-budget strategy, and when the budget is high it is beneficial to use a pure high-budget strategy. The transition area, corresponding to segment $B_{low} \leq B \leq B_{high}$, is typically rather short (see Figs. 4 and 6).

## 2.5 Discussion: motivation from the theoretical analysis

The theoretical framework presented here relies on a simplified model, enabling a rigorous analysis with closed-form solutions. While these solutions may not be directly applicable in real-world deep learning scenarios, they serve as a solid motivation for the SelectAL approach. Empirical evidence demonstrates their practical effectiveness.

In Section 2.2, we introduced a derivative-based test, demonstrating that measuring the model's error function on small data perturbations suffices to determine the appropriate strategy for each budget. This concept serves as the fundamental principle of SelectAL, a method that chooses an active learning strategy by analyzing the model's error function in response to data perturbations.

In Section 2.3, we showed that while the optimal mixed AL strategy can be determined analytically, a practical approximation can be achieved by choosing a single pure AL strategy for each round in active learning. This approach significantly simplifies SelectAL, as it doesn't require the creation of a distinct mixed AL strategy tailored separately to each budget. Instead, SelectAL selects a single pure AL strategy for each active learning round, from a set of existing "pure" AL strategies. In Fig. 5, we observe that this selection consistently outperforms individual pure strategies, confirming our analytical predictions. We also explored various combinations of existing AL strategies, none of which showed a significant improvement over SelectAL, further reinforcing the relevance of predictions of the theoretical analysis.

## 3 SelectAL: automatic selection of active learning strategy

In this section, we present SelectAL – a method for automatically selecting between different AL strategies in advance, by estimating dynamically the relative budget size of the problem at hand. The decision whether $B$ should be considered "large" or "small" builds on the insights gained from the theoretical analysis presented in section 2. The suggested approach approximates the derivative-based test suggested in (1), resulting in a variation of the *attainable optimal mixed strategy* (3).

SelectAL consists of two steps. The first step involves using a version of the derivative-based rule from Section 2.2 to assess whether the current budget $B$ for the given problem should be classified as "high" ($B \geq B_{high}$) or "low" ($B \leq B_{low}$). This is accomplished by generating small perturbations of $\mathbb{L}$ based on both low and high-budget strategies, and then predicting if the current quantity of labeled examples is adequate to qualify as a high-budget or low-budget.

In the second step, we select the most competitive AL strategy from the relevant domain and use it to select the active set $\mathbb{A}$. This approach not only ensures good performance within the specified budget constraints but also provides robustness across a wide range of budget scenarios. It is scalable in its ability to incorporate any future development in active learning methods.

### 3.1 Deciding on the suitable budget regime

In the first step, SelectAL must determine whether the current labeled set $\mathbb{L}$ should be considered low-budget ($B \leq B_{low}$) or high-budget ($B \geq B_{high}$) for the problem at hand. The challenge is to make this decision **without querying additional labels from $\mathbb{U}$**.

To accomplish this, SelectAL requires access to a set of active learning strategies for both low and high-budget scenarios. We denote such strategies by $S'_{low}$ and $S'_{high}$, respectively. Additionally, a random selection strategy, denoted by $S_{rand}$, is also considered.

To determine whether the current budget is high or low, SelectAL utilizes a surrogate test. Instead of requiring additional labels, we compute the result of the test with a derivative-like approach. Specifically, for each respective strategy separately, we remove a small set of points (size $\epsilon > 0$) from $\mathbb{L}$, and compare the reduction in generalization error to the removal of $\epsilon$ randomly chosen points. To implement this procedure, we remove the active sets chosen by either $S'_{low}$ or $S'_{high}$, when trained using the original labeled set $\mathbb{L}$ as the unlabeled set and an empty set as the labeled set.

The proposed surrogate test presents a new challenge: in most active learning strategies, particularly those suitable for high budgets, the outcome relies on a learner that is trained on a non-empty labeled set. To overcome this, we restrict the choice of active learning strategies $S'_{low}$ and $S'_{high}$ to methods that rely only on the unlabeled set $\mathbb{U}$. **This added complexity is crucial**: using the labels of set $\mathbb{L}$ by either $S'_{low}$ or $S'_{high}$ results in a bias that underestimates the cost of removing known points, as is demonstrated in our empirical study (see Fig. 7). No further restriction on $S'_{low}$ and $S'_{high}$ is needed. Empirically, we evaluated several options for $S'_{low}$ and $S'_{high}$, including TypiClust and ProbCover for $S'_{low}$ and inverse-TypiClust and CoreSet for $S'_{high}$, with no qualitative difference in the results.

Our final method can be summarized as follows (see Alg. 1): We generate three subsets from the original labeled set $\mathbb{L}$: $data_{low}$, $data_{high}$ and $data_{rand}$. Each subset is obtained by asking the corresponding strategy $- S'_{low}$, $S'_{high}$, and $S_{rand}$ $-$ to choose a class-balanced subset of $\epsilon \cdot |\mathbb{L}|$ examples from $\mathbb{L}$. Note that this is unlike their original use, as AL strategies are intended to select queries from the unlabeled set $\mathbb{U}$. Importantly, since set $\mathbb{L}$ is labeled, we can guarantee that the selected subset is class-balanced. Subsequently, the selected set is removed from $\mathbb{L}$, and a separate learner is trained on each of the 3 subsets (repeating the process multiple times for $S_{rand}$). Finally, we evaluate the accuracy of each method using cross-validation, employing $1\%$ of the training data as a validation set in multiple repetitions. The subset with the lowest accuracy indicates that the subset lacks examples that are most critical for learning. This suggests that **the corresponding strategy queries examples that have the highest impact on performance**.

---

**Algorithm 1** SelectAL

1: **Input:** $S'_{high}$, $S'_{low}$, $S_{high}$, $S_{low}$, $S_{rand}$, labeled data $\mathbb{L}$, $\epsilon > 0$
2: **Output:** AL strategy
3: $c \leftarrow \max\{\left\lfloor \frac{\epsilon}{\# \text{ of classes}} \right\rfloor, 1\}$
4: **for** $i \in \{$low, high, rand$\}$ **do**
5:     removed_examples $\leftarrow$ query c examples from each class from $\mathbb{L}$ using $S'_i$
6:     $data_i \leftarrow \mathbb{L} \setminus$ removed_examples
7:     $acc_i \leftarrow$ model's accuracy on $data_i$
8: **end for**
9: strategy $\leftarrow \underset{i \in \{low, high, rand\}}{\arg\min} (acc_i)$
10: **return** $S_{strategy}$

---

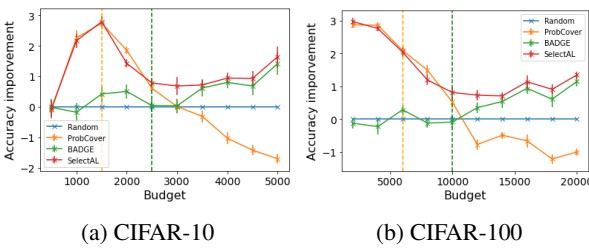

(a) CIFAR-10        (b) CIFAR-100

Figure 5: Accuracy improvement, relative to no active learning (random sampling), for 10 rounds of different AL strategies. We see that while some methods work well only in low budgets and some in high budgets, SelectAL works well in all budgets. Before the first dashed line, SelectAL picked a low-budget strategy, between the dashed lined it picked a random strategy, and after the second dashed line it picked a high-budget strategy. In both experiments, SelectAL picked $S_{low}$ for the first 3 iterations, followed by $S_{rand}$ for 1 iteration and then $S_{high}$.

## 3.2 Selecting the active learning strategy

In its second step, SelectAL selects between two active learning strategies, $S_{low}$, and $S_{high}$, which are known to be beneficial in the low and high-budget regimes respectively. Unlike $S'_{low}$ and $S'_{high}$, there are no restrictions on $\{S_{low}, S_{high}\}$, from which Alg. 1 selects, and it is beneficial to select the SOTA active learning strategy for the chosen domain.

Somewhat counter-intuitively, SelectAL is likely to use different pairs of AL strategies, one pair to determine the budget regime, and possible a different one for the actual query selection. This is the case because in its first step, $S'_{low}$ and $S'_{high}$ are constrained to use only the unlabeled set $\mathbb{U}$, which may eliminate from consideration the most competitive strategies. In contrast, and in order to achieve the best results, $S_{low}$ and $S_{high}$ are chosen to be the most competitive AL strategy in each domain. This flexibility is permitted because both our theoretical analysis in Section 2 and our empirical analysis in Section 4 indicate that the transition points $B_{low}$ and $B_{high}$ are likely to be universal, or approximately so, across different strategies. This is especially important in the high-budget regime, where the most competitive strategies often rely on both $\mathbb{L}$ and $\mathbb{U}$.

## 4 Empirical results

We now describe the results of an extensive empirical evaluation of SelectAL. After a suitable AL strategy is selected by Alg. 1, it is used to query $\mathbb{A}$ unlabeled examples. The training of the deep model then proceeds as is customary, using all the available labels in $\mathbb{A} \cup \mathbb{L}$.

### 4.1 Methodology

Our experimental framework is built on the codebase of [28], which allows for fair and robust comparison of different active learning strategies. While the ResNet-18 architecture used in our experiments may not achieve state-of-the-art results on CIFAR and ImageNet, it provides a suitable platform to evaluate the effectiveness of active learning strategies in a competitive environment, where these strategies have been shown to be beneficial. In the following experiments, we trained ResNet-18 [18] on CIFAR-10, CIFAR-100 [23] and ImageNet-50 – a subset of ImageNet [7] containing 50 classes as done in [38]. We use the same hyper-parameters as in [28], as detailed in Suppl. B.

SelectAL requires two types of active learning strategies: restricted strategies that use only the unlabeled set $\mathbb{U}$ for training, and unrestricted competitive strategies that can use both $\mathbb{L}$ and $\mathbb{U}$ (see discussion above). Among the restricted strategies, we chose *TypiClust* [13] to take the role of $S'_{low}$, and *inverse TypiClust* for $S'_{high}$. In the latter strategy, the most atypical examples are selected. Note that *inverse TypiClust* is an effective strategy for high budgets, while relying solely on the unlabeled set $\mathbb{U}$ (see Suppl. C.1 for details). Among the unrestricted strategies, we chose *ProbCover* [39] for the role of the low-budget strategy $S_{low}$, and *BADGE* [1] for the high-budget strategy $S_{high}$. Other choices yield similar patterns of improvement, as can be verified from Tables 1-2.

In the experiments below, we use several active learning strategies, including *Min margin*, *Max entropy*, *Least confidence*, *DBAL* [10], *CoreSet* [33], *BALD* [22], *BADGE* [1], *TypiClust* [13] and *ProbCover* [39]. When available, we use for each strategy the code provided in [28]. For low-budget strategies, which are not implemented in [28], we use the code from the repository of each paper.

### 4.2 Evaluating the removal of examples with AL in isolation

We isolate the strategy selection test in Alg. 1 as described in Section 3.1. To generate the 3 subsets of labeled examples $\text{data}_{low}$, $\text{data}_{high}$ and $\text{data}_{rand}$, we remove $5\%$ of the labeled data, ensuring that we never remove less than one data point per class. In the low-budget regime, removing examples according to $S'_{low}$ yields worse performance as compared to the removal of random examples, while better performance is seen in the high-budget regime. The opposite behavior is seen when removing examples according to $S'_{high}$. Different choices of strategies for $S'_{low}$ and $S'_{high}$ yield similar qualitative results, see Suppl. C.2.

More specifically, we trained 10 ResNet-18 networks on each of the 3 subsets, for different choices of budget $B$. In Fig. 6, we plot the difference in the mean accuracy of networks trained on $\text{data}_{low}$ and $\text{data}_{high}$, compared to networks trained on $\text{data}_{rand}$, similarly to the proposed test in Alg. 1. In all

the budgets that are smaller than the orange dashed line, SelectAL chooses $S_{low}$. In budgets between the orange and the green dashed lines, SelectAL chooses $S_{rand}$. In budgets larger than the green dashed line, SelectAL chooses $S_{high}$.

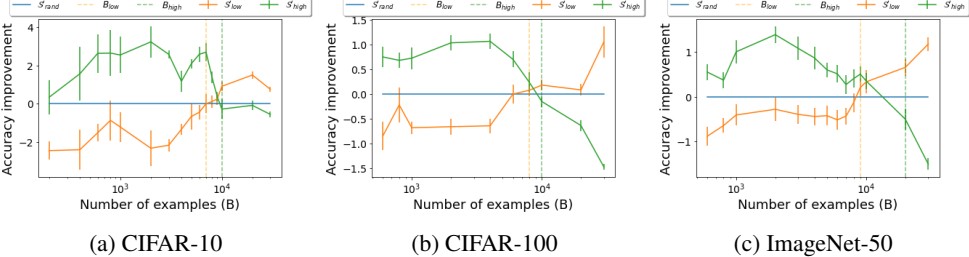

(a) CIFAR-10          (b) CIFAR-100          (c) ImageNet-50

Figure 6: Accuracy gain when using $S'_{low}$ to select points for removal as compared to random selection (orange), or $S'_{high}$ to select points for removal (green). Negative gain implies that the strategy is beneficial, and vice versa.

Table 1: Mean accuracy and standard error of 10 ResNet-18 networks trained on CIFAR-10 and CIFAR-100, using various budgets and active learning strategies. In each dataset, we display results for 3 budget choices: one smaller than $B_{low}$ (left column), one between $B_{low}$ and $B_{high}$ (middle column), and one larger than $B_{high}$ (right column). We highlight in boldface the best result in each column, and additionally all the results that lie within its interval of confidence (the standard error bar). While most strategies are effective only in low or in high budgets, SelectAL is effective in both regimes. As predicted, between $B_{low}$ and $B_{high}$, most AL strategies do not significantly outperform random query selection.

| | CIFAR-10 | | | CIFAR-100 | | |
|---|---|---|---|---|---|---|
| Budget ($\mathbb{L} + \mathbb{A}$) | 100+100 | 7k+1k | 25k+5k | 100+100 | 9k+1k | 30k+7k |
| *random* | $31.8 \pm 0.3$ | $\mathbf{76 \pm 0.3}$ | $87.2 \pm 0.2$ | $5.3 \pm 0.2$ | $\mathbf{39.9 \pm 0.3}$ | $60.7 \pm 0.3$ |
| *TypiClust* | $34.1 \pm 0.4$ | $75.7 \pm 0.3$ | $87.1 \pm 0.2$ | $7.1 \pm 0.1$ | $\mathbf{39.7 \pm 0.4}$ | $60.4 \pm 0.2$ |
| *BADGE* | $31.3 \pm 0.4$ | $\mathbf{76.5 \pm 0.4}$ | $\mathbf{88.1 \pm 0.1}$ | $5.3 \pm 0.2$ | $\mathbf{39.5 \pm 0.3}$ | $\mathbf{61.9 \pm 0.1}$ |
| *DBAL* | $30.2 \pm 0.3$ | $\mathbf{76.4 \pm 0.4}$ | $87.8 \pm 0.1$ | $4.6 \pm 0.2$ | $38.9 \pm 0.4$ | $61.5 \pm 0.2$ |
| *BALD* | $30.8 \pm 0.3$ | $\mathbf{76.3 \pm 0.2}$ | $\mathbf{88 \pm 0.2}$ | $4.9 \pm 0.2$ | $\mathbf{39.8 \pm 0.5}$ | $61.5 \pm 0.2$ |
| *CoreSet* | $29.4 \pm 0.4$ | $75.8 \pm 0.3$ | $87.7 \pm 0.2$ | $5.6 \pm 0.4$ | $39.1 \pm 0.3$ | $61.4 \pm 0.2$ |
| *ProbCover* | $\mathbf{35.1 \pm 0.3}$ | $76.1 \pm 0.3$ | $87.1 \pm 0.1$ | $\mathbf{8.2 \pm 0.1}$ | $\mathbf{40 \pm 0.4}$ | $61.4 \pm 0.3$ |
| *Min Margin* | $30.7 \pm 0.4$ | $71.1 \pm 0.2$ | $\mathbf{87.9 \pm 0.2}$ | $5.2 \pm 0.1$ | $39.2 \pm 0.2$ | $61.6 \pm 0.3$ |
| *Max Entropy* | $30.2 \pm 0.3$ | $76.1 \pm 0.3$ | $87.8 \pm 0.2$ | $4.9 \pm 0.2$ | $39 \pm 0.2$ | $61.6 \pm 0.2$ |
| *Least Confidence* | $29.7 \pm 0.2$ | $\mathbf{76.1 \pm 0.3}$ | $\mathbf{88.1 \pm 0.2}$ | $4.7 \pm 0.4$ | $38.9 \pm 0.4$ | $61.5 \pm 0.2$ |
| *SelectAL* | $\mathbf{35.1 \pm 0.3}$ | $\mathbf{76 \pm 0.3}$ | $\mathbf{88.1 \pm 0.1}$ | $\mathbf{8.2 \pm 0.1}$ | $\mathbf{39.9 \pm 0.3}$ | $\mathbf{61.9 \pm 0.1}$ |

## 4.3   SelectAL: results

In Fig. 5, we present the average accuracy of a series of experiments involving 10 ResNet-18 networks trained over 10 consecutive AL rounds using three AL strategies: $S_{low}$ (ProbCover), $S_{high}$ (BADGE), and our proposed method SelectAL. We compare the accuracy improvement of each strategy to training without any AL strategy. In each AL round, SelectAL selects the appropriate AL strategy based on Alg. 1. Specifically, SelectAL selects $S_{low}$ when the budget is below the orange dashed line, $S_{rand}$ when the budget is between the orange and green dashed lines, and $S_{high}$ when the budget is above the green dashed line. We observe that while $S_{low}$ and $S_{high}$ are effective only for specific budgets, SelectAL performs well across all budgets. In all the experiments we performed, across all datasets, we always observed the monotonic behavior predicted by the theory – SelectAL picks $S_{low}$ for several AL iterations, followed by $S_{rand}$ and then $S_{high}$.

It is important to note that, unlike Fig. 6, where the data distribution of set $\mathbb{L}$ is sampled from the original distribution $\mathcal{D}$ because we analyze the first round (see Section 2.3), in the current experiments the distribution is unknown apriori – $\mathbb{L}$ in each iteration is conditioned on the results of previous iterations, and is therefore effectively a combination of $S_{low}$, $S_{high}$, and $S_{rand}$. Consequently, the transition point determined automatically by Alg. 1 occurs earlier than the one detected in Fig. 6. Notably, while SelectAL chooses an existing AL strategy at each iteration, the resulting strategy outperforms each of the individual AL strategies when trained alone.

In Tables 1-2, we show the performance of SelectAL in comparison with the performance of the baselines (Section 4.1). In all these experiments, SelectAL succeeds to identify a suitable budget regime. As a result, it works well both in the low and high-budget regimes, matching or surpassing both the low and high-budget strategies at all budgets. Note that as SelectAL chooses an active learning strategy dynamically for each budget, any state-of-the-art improvements for either low or high budgets AL strategies can be readily incorporated into SelectAL.

Table 2: Same as Table 1, mean and standard error of 10 ResNet-18 networks trained on ImageNet 50 using different AL strategies, at low, medium, and high budgets.

| B ($\mathbb{L} + \mathbb{A}$) | ImageNet-50 | | |
|---|---|---|---|
| | 100+100 | 7k+1k | 25k+5k |
| *random* | $9.3 \pm 0.2$ | $\mathbf{61.8 \pm 0.4}$ | $79.8 \pm 0.2$ |
| *TypiClust* | $\mathbf{11.3 \pm 0.3}$ | $61.8 \pm 0.5$ | $80.1 \pm 0.2$ |
| *BADGE* | $9.4 \pm 0.2$ | $61.3 \pm 0.5$ | $\mathbf{80.8 \pm 0.2}$ |
| *DBAL* | $9 \pm 0.4$ | $61.1 \pm 0.6$ | $80.1 \pm 0.2$ |
| *BALD* | $9.4 \pm 0.4$ | $61.7 \pm 0.3$ | $\mathbf{80.7 \pm 0.1}$ |
| *CoreSet* | $8.6 \pm 0.3$ | $61.7 \pm 0.2$ | $\mathbf{80.7 \pm 0.3}$ |
| *ProbCover* | $\mathbf{11.4 \pm 0.6}$ | $61.6 \pm 0.8$ | $77.7 \pm 0.3$ |
| *Min Margin* | $9.8 \pm 0.2$ | $61.2 \pm 0.5$ | $80.4 \pm 0.1$ |
| *Max Entropy* | $8.9 \pm 0.2$ | $\mathbf{61.8 \pm 0.4}$ | $80.1 \pm 0.1$ |
| *Least Conf.* | $8.9 \pm 0.1$ | $61.5 \pm 0.4$ | $79.6 \pm 0.5$ |
| *SelectAL* | $\mathbf{11.4 \pm 0.6}$ | $\mathbf{61.8 \pm 0.4}$ | $\mathbf{80.8 \pm 0.2}$ |

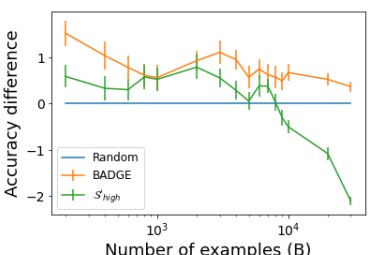

Figure 7: To assess the suitability of BADGE as an example removal strategy $S'_{high}$, we compare it with the original $S'_{high}$ approach whose performance is reported in Fig. 6a, for CIFAR-10. Unlike the original $S'_{high}$ (depicted in green), BADGE (depicted in orange) exhibits a lack of a distinct transition point. Consequently, BADGE is not well-suited as a choice for $S'_{high}$.

**Why $S'_{low}$ and $S'_{high}$ are Restricted?** As discussed in Section 3.1, we have made a deliberate decision to exclude strategies that rely on the labeled set $\mathbb{L}$ while deciding which family of strategies is more suited to the current budget. We now demonstrate what happens when the selection is not restricted in this manner, and in particular, if $S'_{high}$ is chosen to be a competitive AL strategy that relies on the labeled set $\mathbb{L}$ for its successful outcome. Specifically, we repeat the experiments whose results are reported in Fig. 6a, but where strategy $S'_{high}$ – the one used for the removal of examples – is BADGE. Results are shown in Fig. 7. Unlike Fig. 6a, there is no transition point, as it is always beneficial to remove examples selected by BADGE rather than random examples. This may occur because the added value of all points used for training diminishes after training is completed.

**Computational time of SelectAL** SelectAL entails the training of three active learning strategies on a small dataset. Consequently, the computational duration of SelectAL is contingent upon the user's strategy choices. It is worth noting that, in practice, the additional computation time is negligible in comparison to conventional AL techniques. This is due to the fact that $S'_{low}$ and $S'_{high}$ are exclusively trained on $\mathbb{L}$, which is typically significantly smaller than $\mathbb{U}$ - the set used by pure AL strategies. To further expedite computation, one might contemplate training on data perturbations using a more compact model, an issues which we defer the to future research.

## 5 Summary and discussion

We introduce SelectAL, a novel method for selecting active learning strategies that perform well for all training budgets, low and high. We demonstrate the effectiveness of SelectAL through a combination of theoretical analysis and empirical evaluation, showing that it achieves competitive results across a wide range of budgets and datasets. Our main contribution lies in the introduction of the first budget-aware active learning strategy. Until now, selecting the most appropriate active learning strategy given some data was left to the practitioner. Knowing which active learning strategy is best suited for the data can be calculated after all the data is labeled, but predicting this in advance is a difficult problem. SelectAL offers a solution to this challenge, by determining beforehand which active learning strategy should be used without using any labeled data.

**Acknowledgement** This work was supported by grants from the Israeli Council of Higher Education and the Gatsby Charitable Foundations.

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
