# Supplementary

## A Derivation of transition points

Recall that the mean generalization error of mixed strategy $q$ is:

$$E_{\mathcal{L}}(\mathbb{T}) = p \cdot E(qB) + (1 - p) \cdot E(\alpha(1 - q)B).$$

for the differentiable function $E$. The mixture coefficient $q$, which obtains the minimal generalization error, must satisfy

$$0 = \frac{\partial E_{\mathcal{L}}(\mathbb{T})}{\partial q} = pBE'(qB) - (1 - p)\alpha BE'(\alpha(1 - q)B)$$

$$\implies \quad \frac{E'(qB)}{E'(\alpha(1 - q)B)} = \frac{\alpha(1 - p)}{p} \tag{5}$$

In Eq. 5, with specific values for $p$ and $q$, solving for $B$ can determine the budget at which selecting $qB$ examples from $\mathcal{D}_{low}$ yields the best generalization error. By setting $p = q$, we find a budget $B_{eq}$, where the optimal strategy is to keep the distribution unchanged, namely, active learning is not needed. Any smaller budget would necessitate a low-budget strategy, while a higher budget would require a high-budget strategy.

In practice, we only control the source from which the active set of examples $\mathbb{A}$ is sampled, but not the source of the remaining $B - |\mathbb{A}|$ examples. Sampling from $\mathcal{D}_{low}$ all the examples in $|\mathbb{A}|$, results in a total of $\left(p + \frac{|\mathbb{A}|(1-p)}{B}\right) B$ examples being sampled from $\mathcal{D}_{low}$ overall. Plugging such $q$ in Eq. 5, we obtain the maximal budget $B$ for which this strategy is favorable. We refer to this budget as $B_{low}$. Similarly, the concept applies to $B_{high}$, which defined the smallest budget for which sampling $\mathbb{A}$ from $\mathcal{D}_{high}$ is optimal.

Formally, the transition points can now be defined as follows:

- $B_{eq}$ is obtained by solving (5) with $q = p$.
- $B_{low}$ is obtained by solving (5) with $q = p + \frac{|\mathbb{A}|(1-p)}{B}$.
- $B_{high}$ is obtained by solving (5) with $q = p - \frac{|\mathbb{A}|(1-p)}{B}$.

## B Hyper-parameters

### B.1 Supervised training

When training on CIFAR-10 and CIFAR-100, we used a ResNet-18 trained over 50 epochs. We used an SGD optimizer, with 0.9 Nesterov momentum, 0.0003 weight decay, cosine learning rate scheduling with a base learning rate of 0.025, and batch size of 100 examples. We used random croppings and horizontal flips for augmentations. An example use of these parameters can be found at [28].

When training ImageNet-50, we used the same hyper-parameters as CIFAR-10/100, only changing the base learning rate to 0.01 and the batch size to 50.

### B.2 Unsupervised representation learning

**CIFAR-10/100** We trained SimCLR using the code provided by [38] for CIFAR-10 and CIFAR-100. Specifically, we used ResNet18 with an MLP projection layer to a 128 vector, trained for 500 epochs. All the training hyper-parameters were identical to those used by SCAN. After training, we used the 512 dimensional penultimate layer as the representation space. As in SCAN, we used an SGD optimizer with 0.9 momentum and an initial learning rate of 0.4 with a cosine scheduler. The batch size was 512 and a weight decay of 0.0001. The augmentations were random resized crops, random horizontal flips, color jittering, and random grayscaling. We refer to [38] for additional details. We used the L2 normalized penultimate layer as embedding.

**ImageNet-50** We extracted embedding from the official (ViT-S/16) DINO weights pre-trained on ImageNet. We used the L2 normalized penultimate layer as embedding.

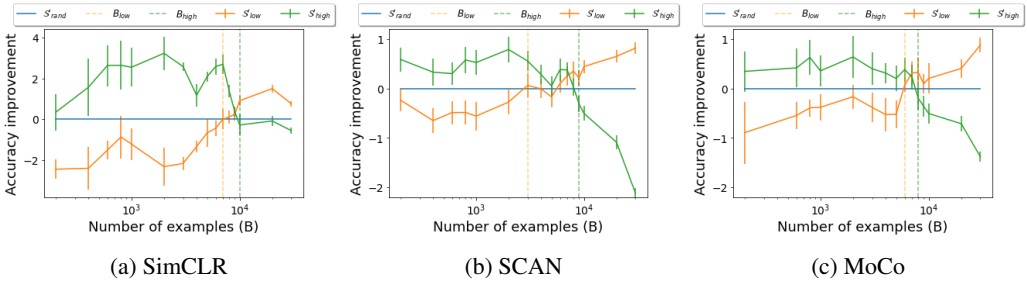

(a) SimCLR      (b) SCAN      (c) MoCo

Figure 8: Similar to Fig. 6, but removing examples according to different feature spaces in CIFAR-10. Accuracy gain when using $S'_{low}$ to select points for removal as compared to random selection (orange), or $S'_{high}$ to select points for removal (green). Negative gain implies that the strategy is beneficial, and vice versa.

# C   Additional experimental results

## C.1   High budget strategies

In Section. 4, we are required to use a high budget strategy $S'_{high}$, which relies in its computation only on the unlabeled set $\mathbb{U}$. We use *inverse-TypiClust*, which is calculated similarly to *TypiClust*, only using the most atypical example at each iteration instead of the most typical example. In Fig. 10, we plot the performance of such a strategy on CIFAR-10, as a function of budget $B$, similarly to the analysis in Fig. 4.

We see that while *inverse-TypiClust* is not a competitive high-budget strategy, it still outperforms random sampling in the high-budget regime, making it a suitable AL strategy for this regime.

## C.2   Different choices for low and high budget strategies in the decision process

In Fig. 6, we plot the accuracy of TypiClust as $S'_{low}$ and Inverse-Typicluse as $S'_{high}$, under different budgets against training without active learning. SelectAL utilizes these strategies for selection. When $S'high$ outperforms random and $S'low$ underperforms, SelectAL opts for a low-budget strategy. Conversely, if $S'high$ underperforms while $S'low$ outperforms random, a high-budget strategy is chosen.

In Fig. 9, we plot the same experiment as done in Fig. 6, choosing BADGE as $S'_{high}$ and ProbCover as $S'_{low}$. We see that while the accuracy improvements of these strategies may differ, the decision made by SelectAL remains the same: SelectAL performs the same selections in every given budget. These results suggest that SelectAL works with a variety of underlying strategies.

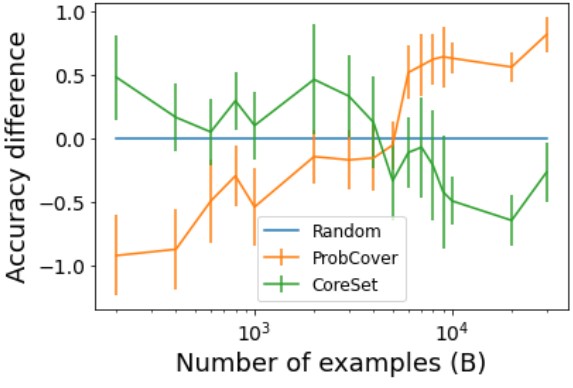

Figure 9: Similar to Fig. 6, using BADGE as $S'_{high}$ and ProbCover as $S'_{low}$.

### C.3 Other feature spaces

#### C.3.1 Other feature spaces: removing data

In section 3.2, we propose an active learning method that determines the budget size by removing examples in a given feature space. The feature space used in section 3.2 was obtained by SimCLR, as these features proved beneficial to several low-budget active learning methods.

In this section, we check the dependency of MiSAL on the specific choice of feature space. In Fig. 8, we plot the strategy selection test as described in Alg. 1 in Section 3.1. The plotted results are trained on CIFAR-10. In order to generate the 3 subsets of labeled examples $data_l$, $data_h$ and $data_r$, we remove $5\%$ of the labeled data (but not less than 1 datapoint per class). This test is done using 3 different feature spaces 1. MoCo [17], a transformer based approach. 2. SimCLR, as done in section 3.2. 3. SCAN [38].

Similarly to the results reported in section 3.2, we can see that using any of the 3 feature spaces resulted in a similar result – MiSAL would behave similarly regardless of the choice of the underlying feature space.

#### C.3.2 Other feature spaces in TypiClust

In Table 1 and Table 2, we plot the results of different AL strategies across different datasets and budgets. Low-budget strategies such as TypiClust and ProbCover require the choice of feature space to work properly. Following the original papers, we used the feature space given by SimCLR trained on the entire unlabeled pool $\mathbb{U}$.

To check whether the choice of the feature space affects the results of the low-budget performance, we trained TypiClust on the TinyImageNet with various choices of feature spaces.

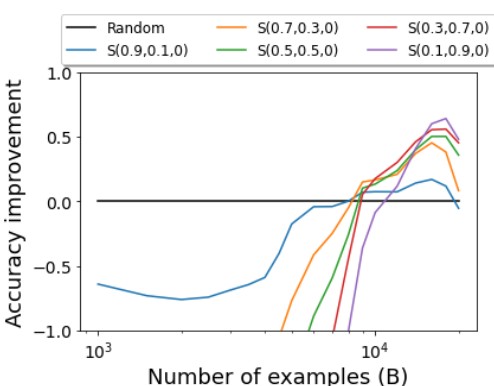

Figure 10: Accuracy gain by *inverse-TypiClust*, as compared to random query sampling.

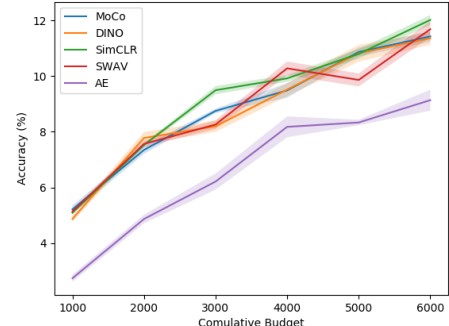

Figure 11: Comparing TypiClust with different representations on TinyImageNet for 5 AL iterations in the low budget regime. The active set size is $|\mathbb{A}| = 1000$. The final test accuracy in each iteration is reported. The shaded area reflects standard error. All results are with respect to the 'random' representation, which is the pixel value of each image.

In Fig. 11, we plot 5 active learning iterations with an active set of $|\mathbb{A}| = 1000$ of ResNet-50 trained on TinyImageNet. We considered 5 different feature spaces: 1. MoCo [17], a transformer based approach. 2. DINO [3], an SSL-based approach. 3. SimCLR, which was used in the original TypiClust paper. 4. SWAV an SSL-based approach. 5. A simple autoencoder on the pixel values (AE). We found that except for the AE, all methods perform similarly, suggesting that the choice of the representation space has little effect on the training of low-budget methods such as TypiClust.