# OpenReview forum: "How to Select Which Active Learning Strategy is Best Suited for Your Specific Problem and Budget"
_NeurIPS.cc/2023/Conference — NeurIPS 2023 poster_

### Official Review · Reviewer_mgMr · 2023-07-07

**Soundness:** 3 good
**Presentation:** 2 fair
**Contribution:** 3 good
**Rating:** 5
**Confidence:** 3

**Summary:**

Some recent works have shown that typical examples are more beneficial than more uncertain ones under low budget setting in active learning. However, there has been no prior works that address how to distinguish whether a budget is in low or high budget regime, or how to determine which approach should be taken.

The authors propose SelectAL, an algorithm that handles the tasks using derivative-based test, for the first time. The algorithm is theoretically and empirically supported in Section 2 and 4, respectively.

**Strengths:**

1. This is the first paper that addresses how to automatically determine selection strategies based on the budget in active learning.
2. The authors provide some theoretical justification for the derivative-based test method.

**Weaknesses:**

1. Although Strength 1 is a great contribution of this paper, there are several assumptions some of which may not be realistic. Intuitively, efficiency and realizability are reasonable assumptions but it would be hard to check if universality is generally true or not. Furthermore, the uniqueness of $B_{eq}$ (line 120-121) seems like a strong assumption.
2. It is somewhat hard to follow the flow of this paper. It is partially because of the complexity of theoretical work, but also because there are many assumptions to justify the theory. But upon the clarification of the questions below, I am willing to increase the score.

**Questions:**

1. I am very confused with the definition of the generalization errors. If my understanding is correct, $E_{low}$ and $E_{high}$ are the generalization errors on the same unseen test set using $\mathcal{L}_{low}$ and $\mathcal{L}_{hight}$, each of which are trained on $\mathcal{D}_{low}$ and $\mathcal{D}_{high}$. How is it related to the equation in line 111?
2. What the equation in line 111 is saying is that the generalization error of $\mathcal{L}$ trained on $B$ examples can be expressed as the convex combination of $E(qB)$ and $E(\alpha(1-q)B)$. But, $E(qB)$ is from a learner trained only on $qB$ examples and $E(\alpha(1-q)B)$ is from the other learner trained only on $(1-q)B$ examples. Is it generally true or even close to the reality? If the generalization error is a function of the test examples given the same learner, I think the convex combination makes sense but since it is a function of training examples, I am not sure it is reasonably true.
3. Where are equalities for q coming from for $B_{low}$ and $B_{high}$ in  Appendix A?
4. If the uniqueness of $B_{eq}$ does not hold, what happens to the Eq.(3), particularly, for the case $|A| \rightarrow 0$?
5. Could the authors provide more details about the experiment setting used to produce Figure 2 such as a linear $\mathcal{L}$, the dataset, and reasoning behind of choosing exponential error function?
6. For selectAL, did the authors observe some cases where {low, high, rand} are selected alternatively unlike the expected monotonic behavior? Can the authors provide the sequence of selections on CIFAR-10 and ImageNet-50?

**Limitations:**

Please refer to the weakness and questions above.

---

> ### Author Rebuttal · Authors · 2023-08-07
>
> Thanks much for the detailed and precise list of questions. We wish to emphasize that some brevity in the manuscript was called for because the general framework is not original - it is adopted with all its assumptions from the work of [1]. This study demonstrated that the model, despite its simplified assumptions, allows for a clear differentiation between low and high budgets, and can further be subjected to rigorous analysis. We note that due to the inherent simplifications in the model, some assumptions are hard to justify in realistic settings and therefore any conclusion derived from the model requires exhaustive empirical testing to validate its relevance in real settings.
>
> Answers to questions:
>
> ---
> 1+2)
>
> The model under analysis consists of two learners, $L_{low}$ and $L_{high}$, trained on separate datasets $D_{low}$ and $D_{high}$, respectively. $L_{low}$ is trained and evaluated solely on $D_{low}$, while $L_{high}$ on $D_{high}$.
>
> We then define a mixture model, combining $L_{low}$ and $L_{high}$. The model is trained and tested on a mixture distribution, where each example is drawn from $L_{low}$ with probability $p$ and from $L_{high}$ with probability $(1-p)$.
>
> The main question such a framework can address is whether changing the training distribution of the combined model, while keeping the test set unchanged, is beneficial. Under certain assumptions, the results suggest that in low-budget scenarios, more examples should come from $D_{low}$, and in high-budget scenarios, more examples should come from $D_{high}$.
>
> Since the distribution of the data is a convex combination of $2$ distributions, so is the error of the combined model, which is depicted in line 111. A combined model that gets $qB$ examples from $D_{low}$ and $(1-q)B$ examples from $D_{high}$ is expected to have a generalization error of $pE(qB) + (1-p)E(\alpha (1-q)B)$ on the mixture distribution.
>
> ---
> 3)
>
> In Eq. 4, with specific values for $p$ and $q$, solving for $B$ can determine the budget at which selecting $qB$ examples from $D_{low}$ yields the best generalization error. By setting $p=q$, we find a budget ($B_{eq}$) where the optimal strategy is to keep the distribution unchanged, namely, active learning is not needed. Any smaller budget would necessitate a low-budget strategy, while a higher budget would require a high-budget strategy.
>
> In practice, we only control the source from which the active set of examples $\mathbb{A}$ is sampled, but not the source of the remaining $B-|\mathbb{A}|$ examples. Sampling from $D_{low}$ all the examples in $|\mathbb{A}|$, results in a total of $\left(p+\frac{\mathbb{A}(1-p)}{B}\right)B$ examples being sampled from $D_{low}$ overall. Plugging such $q$ in Eq. 4, we obtain the maximal budget $B$ for which this strategy is favorable. We refer to this budget as $B_{low}$. Similarly, the concept applies to $B_{high}$, which defined the smallest budget for which sampling $\mathbb{A}$ from $D_{high}$ is optimal.
>
> ---
> 4)
>
> If $B_{eq}$ is not unique, it implies that Eq. 4 have multiple solutions, suggesting the presence of a multitude of low and high-budget regimes. However, in our empirical investigations, we have not observed this phenomenon.
>
> When multiple $B_{low}$ and $B_{high}$ values exist, they delineate distinct budget thresholds. For budgets below the smallest $B_{low}$ value, the best strategy involves sampling the entire active set from $D_{low}$. Conversely, for budgets surpassing the largest $B_{high}$ value, the optimal strategy is to sample the entire active set from $D_{high}$. In the intermediate range between these two thresholds, determining the precise optimal strategy becomes more challenging, and random sampling may be preferred. As $|\mathbb{A}|\leftarrow 0$, the minimum $B_{low}$ approaches the maximum $B_{high}$, resulting in the same optimal strategy in the limit as described in Eq. 3.
>
> ---
> 5)
>
> There is no specific dataset or learner in the experiment in Fig 2. Once the error function is defined, datasets and learners are not explicitly required -- the analysis is done based on the error function alone.
>
> The reason we chose the exponential form for the error function is twofold. Firstly, when examining the error functions of neural networks on real datasets, they often exhibit behavior that resembles an exponential pattern. Thus, using an exponential form in this example allows us to analyze a scenario that aligns with the behavior of empirical error functions in realistic scenarios. Secondly, we opted to utilize the same example from [1], as it was a recognized and relevant illustration, ensuring continuity with previous art.
>
> ---
> 6)
>
> In all our experiments and across various settings, we consistently observed a monotonic pattern in SelectAL's behavior. Specifically, SelectAL followed a sequence of low-budget strategy for several rounds, then a random strategy for another set of rounds, then a high-budget strategy for the remaining rounds.
>
> For both CIFAR-10 and CIFAR-100 datasets, as depicted in Fig 5, the selection sequence of SelectAL was as follows: It initially employed the low-budget strategy for 3 iterations. then random strategy for 1 iteration and rom that point onwards, it favored the high-budget strategy. The specific values for these iterations are highlighted in Fig 5, using orange and green dashed lines.
>
> ---
> In postfix, we sincerely wish to thank you for your valuable feedback. This thorough review of our work and the insightful questions you raised help us recognize the areas that need further clarification, strengthening our work.
>
> In the camera-ready revision, we will carefully address each of the concerns you have raised, aiming to enhance the overall coherence and comprehensibility of the manuscript. We are committed to incorporating your feedback to deliver a more polished and reader-friendly paper.
>
> ---
> [1] Hacohen, Guy, et al. "Active Learning on a Budget: Opposite Strategies Suit High and Low Budgets." ICML 2022.

---

### Official Review · Reviewer_5Syc · 2023-07-07

**Soundness:** 3 good
**Presentation:** 2 fair
**Contribution:** 3 good
**Rating:** 5
**Confidence:** 4

**Summary:**

This paper considers the selection of active learning methods under different budgets. Previous studies have shown that different active learning methods perform differently under different labeling budgets, and incorrect selection can lead to model performance inferior to that of random sampling baselines. To address this issue, the authors conducted theoretical analysis and proposed the SelectAL method, and related experiments showed that this method can achieve consistent performance improvements under different budgets.

**Strengths:**

1. The idea presented is interesting and it tackles an important issue in active learning.
2. The experimental results indicate that the proposal consistently improved performance, supporting the claims made in the paper.

**Weaknesses:**

1. There is room for improvement in the technological representation.
2. The connection between theoretical results and practical methods seems to be weak.
Miscellaneous: The text font in the figures should be kept the same as the main body.

**Questions:**

1. What is the physical meaning of $E_{low}(x)=E_{high}(\alpha x)$ in the assumptions (Section 2.1)? Can you provide a more intuitive explanation?
2. The theoretical results (Section 2) assume that there is a unique and minimal solution for Eq.(1). What kind of E satisfies this assumption, and is this condition general enough to support practical applications?
3. The proposed method (Section 3) seems weakly connected to the theoretical analysis. The theoretical results show that under specific assumptions, there exist sampling boundaries B_low and B_high for the Low-Budget Method and High-Budget Method, which can be calculated through closed-form solutions. However, the SelectAL method used in the proposed method relies on a training-evaluating method to identify the type of samples that are currently lacking. Can you provide more connections between theoretical results and practical methods?
4. In the experiments (Figure 6), there are indeed performance differences between different AL methods, but the differences are small (≤2%). Have you observed more severe phenomena? In Tables 1 and 2, there are similar phenomena, and the performance differences between all methods are not significant. Specifically, in the low-budget scenario, the classification performance is so poor that the model cannot be deployed. Is it necessary to introduce complex methods to deal with this? Perhaps, random sampling followed by the consideration of high-budget methods is already sufficient.

---
Thanks for the detailed clarifications, which have addressed my concerns. I would like to keep my score.

**Limitations:**

The authors do not provide a discussion about the limitations and potential negative societal impact.

---

> ### Author Rebuttal · Authors · 2023-08-07
>
> **$B_{low}$ and $B_{high}$, and the connection between the theoretical analysis and the practical methods**
>
> Indeed, the theoretical framework presented in this study relies on a simplified model. This makes possible a rigorous analysis and the derivation of closed-form solutions, unlike the involved analysis of real-world deep network training scenarios. While these closed-form expressions may be irrelevant in real deep learning scenarios, approximating $B_{low}$ and $B_{high}$ plays a vital role in SelectAL. The removal of examples in Algorithm 1 serves the purpose of calculating whether the current budget falls above or below these thresholds, as explained in detail in Figure 6. Overall, our whole approach is greatly motivated by the theoretical results; in effect, the main algorithm puts the theoretical results into actual use by empirically evaluating the thresholds and boundaries identified in the analysis.
>
> The following discussion may clarify further connections between the theoretical results and the actual method. We note that the proposed theory conveys two main such insights, which are implemented and tested in our suggested method:
>
> 1. The effectiveness of a derivative-like test, highlighted in Section 2.2 and emphasized in Equation 1, proves to be valuable in determining the budget that best suits the problem at hand. This theoretical result is put into action in the suggested algorithm, which perturbs the given labeled set to discern the appropriate budget regime.
>
> 2. The theoretical analysis in Section 2.3 demonstrates that choosing a mixture of active learning (AL) strategies in each AL round can be well approximated by selecting a single AL strategy in each round and allowing for the flexibility to switch strategies between rounds. This insight significantly simplifies the algorithm, as it shifts the focus from selecting the best mixture of several strategies to making a binary choice between two distinct strategies in a pure manner. Additionally, as observed in Fig. 5, allowing for a different pure AL strategy in each round outperforms the use of a single pure strategy throughout, validating the predictions made in our analysis.
>
> For the camera-ready version, we will add a comprehensive discussion that explicitly emphasizes these points, thereby reinforcing the cohesion between the theoretical foundations and the practical ingredients of the proposed approach.
>
> ---
> **$E_{low}(x)=E_{high}(\alpha x)$**
>
> The error functions $E_{low}$ and $E_{high}$ measure the generalization error of a classifier as a function of the size of its training data. For example, it could be some exponential decreasing function: the error drops fast if you increase the size of the training data at first, but as more data is added there is some diminishing returns effect.
>
> $E_{low}$ and $E_{high}$ are 2 such functions, for 2 similar classifiers, trained on either an easy dataset, corresponding to the low-budget, or a hard dataset, corresponding to the high budget. The $E_{low}(x)=E_{high}(\alpha x)$ states that the error functions in such case would be from a similar family of functions, i.e if $E_{low}$ decreases exponentially, then $E_{high}$ also does, but with a different rate of decay. This assumption is rather realistic -- measuring the error functions of neural networks on different datasets indeed shows that they decay in an exponential-like manner but with different decay rates.
>
> ---
> **Uniqueness of Eq. 1**
>
> In the case where there are multiple solutions for Eq. 1, the core analysis and its qualitative implications remain unchanged. In such case, we observe the existence of several thresholds $B_{eq}$, each having its distinct $B_{low}$ and $B_{high}$. To maintain consistency in our analysis, we can employ the minimal threshold as $B_{low}$ and the maximal threshold as $B_{high}$, while adopting a random AL strategy within this range. It's important to highlight that in practical scenarios when training deep networks on real datasets, the observed solutions have consistently been unique (see Fig. 6 for a concrete example).
>
> ---
> **Size of the differences observed**
>
> In Tables 1 and 2, we acknowledge that the differences between models trained using active learning (AL) strategies and those trained on random examples might appear relatively small. This is due to the fact that we consider a single round of AL, where only a limited number of examples are chosen by the AL strategy. Consequently, the impact of these few added examples on the overall training process may not be substantial. However, it is important to note that the primary objective in this context is to assess whether there is an improvement or deterioration resulting from the AL approach.
>
> In contrast, in Fig. 5, we demonstrate the results of a multi-round AL experiment, where a larger fraction of examples are selected by AL strategies. In this multi-round setting, the improvements achieved through active learning are significant and noticeable.
>
> Regarding the low-budget results, it is essential to consider that we are dealing with a dataset containing only 200 examples. In such limited data scenarios, achieving substantial improvements beyond what is reported in the tables can be challenging. This closely mirrors real-life situations where data can be sparse, making active learning techniques all the more valuable.
>
> However, we also recognize that there are potential ways to enhance results in low-budget scenarios, especially when unlabeled examples are available, and a semi-supervised approach is employed. In such cases, the separation between low and high-budget regimes has been demonstrated, and the use of SelectAL may be advantageous. While this is a promising avenue, we want to emphasize that our manuscript's main focus is not on semi-supervised scenarios but on showcasing the effectiveness of SelectAL in identifying the budget regime of the problem in advance.

---

> > ### Comment · Reviewer_5Syc · 2023-08-16
> >
> > Thank you for your detailed clarifications, which have addressed my concerns. I would like to keep my score and recommend acceptance of this paper.

---

### Official Review · Reviewer_GcEN · 2023-07-07

**Soundness:** 3 good
**Presentation:** 2 fair
**Contribution:** 3 good
**Rating:** 5
**Confidence:** 4

**Summary:**

The paper studies an algorithm selection problem for active learning under different labeling budgets. A theoretical analysis is conducted The proposed algorithm uses a derivative approach by removing small sets of labeled examples to approximate the effectiveness of different active learning algorithms. Empirical results are shown for different labeling budget that SelectAL chooses the best active learning algorithm. Experiments are conducted on CIFAR-10, CIFAR-100 and ImageNet-50.

**Strengths:**

1. The paper studies an important problem of how to choose the best active learning algorithm for a labeling budget.
2. Experiments demonstrates the effectiveness of the method in specific settings.
3. It is interesting to empirically see that random sampling performs as good as other active learning algorithms when given a moderate number of labeling budget.

**Weaknesses:**

I have major concerns with the writing and structure of the paper.

1. The theoretical analysis and the actual algorithm seem to be very loosely connected. Currently, section 3 contains very little motivation from section 2. Also, the analysis framework makes assumption on universality making it unlikely to reflect the performance in practice.

2. The data removal process from a labeled set is not immediate to the reader but is the key underlying tool for the derivate approximation. How does one remove labeled examples from a labeled set using an active learning algorithm?

3. Throughout the paper, the data removal algorithms are used as TypiClust and inverse-TypiClust. Instead of formulating the algorithm as using general active learning algorithms for removal, the authors may discuss more in-depth why these two algorithms are actually effective in predicting low and high budget scenarios.

Furthermore, the experiments seem to be inconsistent. Specifically, even though CIFAR-10 and CIFAR-100 have the same pool size, the experiments are conducted with inconsistent budgets. For example, 7k+1k and 25k+5k are used for CIFAR-10 while 9k+1k and 30k+7k are used for CIFAR-100. Since the paper's focus is on different labeling budgets, I believe more budget settings are also needed beyond just three.

Lastly, algorithm selection methods have been studied before, but not for different budgets. The paper could benefit from the following related work:
1. Baram, Y., Yaniv, R. E., & Luz, K. (2004). Online choice of active learning algorithms. Journal of Machine Learning Research, 5(Mar), 255-291.
2. Hsu, W. N., & Lin, H. T. (2015, February). Active learning by learning. In Proceedings of the AAAI Conference on Artificial Intelligence (Vol. 29, No. 1).
3. Pang, K., Dong, M., Wu, Y., & Hospedales, T. M. (2018, August). Dynamic ensemble active learning: A non-stationary bandit with expert advice. In 2018 24th International Conference on Pattern Recognition (ICPR) (pp. 2269-2276). IEEE.
4. (Note the first version came out Feb 2023.) Zhang, J., Shao, S., Verma, S., & Nowak, R. (2023). Algorithm Selection for Deep Active Learning with Imbalanced Datasets. arXiv preprint arXiv:2302.07317.


**Questions:**

1. What's the underlying intuition and reasoning of why TypiClust and inverse-TypiClust can correctly predict the budget scenarios.
2. Why are the particular initial seed set size and batch size selected for CIFAR-10 and CIFAR-100.

**Limitations:**

Sufficient.

---

> ### Author Rebuttal · Authors · 2023-08-07
>
> **Connections between the theoretical analysis and the actual algorithm presented:**
>
> The theoretical analysis presented in our work conveys two main take-home messages:
>
>  1. The effectiveness of a derivative-like test, highlighted in Section 2.2 and emphasized in Equation 1, proves to be valuable in determining the budget that best suits the problem at hand. This theoretical result is put into action in the suggested algorithm, which perturbs the given labeled set to discern the appropriate budget regime.
> 2. The theoretical analysis in Section 2.3 demonstrates that choosing a mixture of active learning (AL) strategies in each AL round can be well approximated by selecting a single AL strategy in each round and allowing for the flexibility to switch strategies between rounds. This insight significantly simplifies the algorithm, as it shifts the focus from selecting the best mixture of several strategies to making a binary choice between two distinct strategies in a pure manner. Additionally, as shown in Fig. 5, allowing for a different pure AL strategy in each round outperforms the use of a single pure strategy throughout, validating the predictions made in our analysis.
>
> We understand from the reviews that these connections, between the analysis and the empirical section, could be better elucidated in the current manuscript. Accordingly, for the camera-ready version, we will add a comprehensive discussion that explicitly emphasizes these points, thereby reinforcing the cohesion between the theoretical foundations and the practical ingredients of the proposed approach.
>
> ---
> **Removing data using AL algorithm, and the choice of TypiClust and inverse TypiClust:**
>
> The rationale for choosing the specific methods of TypiClust and inverse-TypiClust is the following:
>
> To remove examples from a labeled set using an active learning algorithm, we must be able to consider the AL algorithm as if it operates with an empty labeled set, denoted by $\mathbb{L}$, while its unlabeled pool $\mathbb{U}$ is composed of the original labeled dataset. This restricts our choice of AL algorithms quite drastically. This is because most (if not all) uncertainty-based algorithms are not  well-defined in such cases, since they typically make use of a classifier trained on $\mathbb{L}$, which in our scenario is empty. In contrast, TypiClust [1] and Inverse-TypiClust present suitable solutions for these settings, as they tackle a covering problem directly on the data itself, independent of any specific classifier trained on $\mathbb{L}$. Additionally, in the original paper it was shown that typiclust is a very good performer when given an empty $\mathbb{L}$.
>
> Based on the above, and the lack of competitive alternative methods that can be effective when $\mathbb{L}$ is empty, we chose TypiClust and inverse-TypiClust. This choice was empirically tested (see Fig. 6), where it was shown to be effective.
>
> While this concept is presently explained in the manuscript in the paragraph starting at line 218, we will emphasize and clarify this point in future revisions, as it is an important part of our suggested algorithm.
>
> Nevertheless, it is essential to highlight that the proposed algorithm is not exclusive to this choice of TypiClust and Inverse-TypiClust. Alternative AL strategies, such as ProbCover [2] as a low-budget option or CoreSet [3] as a high-budget option, can also be effective choices in place of TypiClust and Inverse-TypiClust, respectively. In Fig. 1 in the attached PDF, we report the decisions made by SelectAL when using ProbCover and CoreSet instead of TypiClust and inverse-Typiclust, demonstrating this point. We will add this figure to the appendix, and discuss it in the main paper.
>
> ---
> **Different budget settings for CIFAR-10 and CIFAR-100**
>
> The size of the low-budget regime varies across different tasks. Intuitively, in the low-budget regime, you only have enough examples to provide a coarse description of the underlying problem. As the difficulty of the task increases, the low-budget regime becomes larger. Since CIFAR-100 is a more challenging problem compared to CIFAR-10 due to its tenfold increase in the number of classes, it has a larger low-budget regime despite both datasets sharing the same pool size.
>
> Table 1 in the manuscript showcases specific settings for a single-round active learning scenario, selected to be representative of different budget levels one might encounter. Considering that the low budget in CIFAR-100 is larger, we appropriately adjusted the settings for the mid-budget and high-budget scenarios to reflect this difference.
>
>
> Having said all that, it is important to note that Table 1 is a **showcase**, and by no means shows all the conditions tested. For example, Fig. 5 shows results with many more cases. It summarizes the results of experiments with SelectAL in multi-round AL settings, effectively testing it in a large array of possible budgets, including low, mid, and high budgets, for both CIFAR-10 and CIFAR-100. This specific figure clearly shows that SelectAL works well in many settings, where it outperforms other AL strategies by a significant margin.
>
> ---
> **Related work**
>
> Thank you for bringing this set of interesting papers to our attention. Indeed they are not directly related to our work, as they focus on selecting AL algorithms under different settings than ours. Nevertheless, it will strengthen the paper to discuss our method in a broader context. This will be done for the camera-ready revision.
>
> ---
> [1] Hacohen, Guy, et al. "Active Learning on a Budget: Opposite Strategies Suit High and Low Budgets." International Conference on Machine Learning. PMLR, 2022.
>
> [2] Yehuda, Ofer, et al. "Active learning through a covering lens." Advances in Neural Information Processing Systems 35 (2022): 22354-22367.
>
> [3] Sener, Ozan, and Silvio Savarese. "Active Learning for Convolutional Neural Networks: A Core-Set Approach." International Conference on Learning Representations. 2018.

---

> > ### Comment · Reviewer_GcEN · 2023-08-10
> >
> > Thank you for your clarifications. I believe most of my concerns are addressed. The only problem is in the PDF, the figure does not seem to only contain performance of SelectAL?

---

> > > ### Author Response · Authors · 2023-08-10
> > >
> > > We deeply appreciate your prompt feedback and the opportunity to further clarify our work.
> > >
> > > The figure in the PDF indeed does not show the performance of SelectAL. Instead, the figure is focused solely on illustrating the decision-making process within the SelectAL algorithm, rather than explicitly showcasing its performance enhancement. In a manner analogous to Figure 6 in the original manuscript, this figure demonstrates for each budget the values that the SelectAL will calculate when perturbing the data.
> > >
> > > As this figure was calculated using ProbCover and CoreSet instead of TypiClust and inverse-TypiClust, it shows the decisions that SelectAL will make using these AL strategies instead. Since the qualitative behavior depicted in this figure is the same as the behavior in Fig. 6a in the original manuscript, the resulting SelectAL will perform identically to SelectAL which is based on TypiClust and inverse-Typiclust.
> > >
> > > Therefore, the performance of SelectAL using these underlying strategies will be identical to the performance depicted in Table 1 and Fig 5a, which is already shown to be good.
> > >
> > > Thank you once again for your valuable engagement with our work. We would be more than happy to address any other concerns you may have.

---

> > > > ### Comment · Reviewer_GcEN · 2023-08-16
> > > >
> > > > My concern is the current framing in the paper where you do not frame it as SelectAL only uses TypiClust and inverse Typiclust. Instead, the entire section 3 talks about using a general set of $S_{low}’$ and $S_{high}’$. I think you could be more specific in your algorithm box as the entire paper only uses one such set of low and high budget algorithm. Just to clarify, my concern was not on limited experiments on $S_{low}$ and $S_{high}$, but instead on $S_{low}’$ and $S_{high}’$.
> > > >
> > > > Nevertheless, I believe the most major concern of mine has been addressed and will raise my score to 5.

---

### Official Review · Reviewer_uJGw · 2023-07-09

**Soundness:** 3 good
**Presentation:** 2 fair
**Contribution:** 2 fair
**Rating:** 6
**Confidence:** 4

**Summary:**

The literature has shown that the optimal type of active learning (AL) strategy depends on the type of budget. Methods based on uncertainty sampling are most effective when the budget is large, and methods based on typicality work better in the low budget setting. However, what qualifies "large" or "low" depends on different criteria such as the data set and its size, the type of neural network architecture that is trained etc.
The paper proposes a general approach that automatically determines whether a given setting is low or high budget by decoupling the data set in different parts. The method is divided in two parts. In the first part, a different active learning strategy is used depending on whether a labeled image belongs to the low or high budget setting. In the second part, unlabeled images are added to the labeled set.

**Strengths:**

Quality and clarity: The general motivation of the approach (i.e. combining low and high budget settings and using different strategies for each) is clear and makes sense given the recent AL literature. The first part of the algorithm is clear and more or less described in Algorithm 1. The clarity of the second part could be improved by writing its pseudo-code (in the main paper or appendix).

Originality: Similar ideas that consider different strategies for different parts of the data set were recently published in the literature (e.g. [A,B]).
Jain et al. [A] show that low-budget settings follow a linear law whereas high-budget settings follow a power law in the standard classification setting. They also try to estimate whether a setting is low or high budget.
Mahmood et al. [B] also consider to learn the optimal budget over multiple datasets (in the multi-variate case) to improve model performance while minimizing labeling cost. This is related to the case where each dataset corresponds to a type of budget.
However, [A,B] do not consider active learning strategies.

[A] Jain et al., A Meta-Learning Approach to Predicting Performance and Data Requirements, CVPR 2023
[B] Mahmood et al., Optimizing data collection for machine learning, NeurIPS 2022

Significance: The idea of the paper is interesting as it allows to automatically find the optimal AL strategy and type of budget. However, I wonder how much it would be useful in real-world applications since it seems much more expensive to run than standard AL approaches (by constantly checking how much samples improve the performance, and then to determine whether they have to be added to the low or high budget).

**Weaknesses:**

- As mentioned above, the whole method should be clearly written in the main paper or appendix to improve clarity.

- Is the method scalable to real-world applications? If so, isn't it cheaper to select a few uncertainty strategies (in large scale applications) to optimize the general labeling cost?

- Due to its nature, the proposed method SelectAL returns the same scores as the best baselines (TypiClust and ProbCover, according to Table 1 and 2). Would there be a way to combine low and high budget strategies to obtain significantly better performance than a single baseline?

**Questions:**

See above

---

> ### Author Rebuttal · Authors · 2023-08-07
>
> We thank the reviewer for the references; indeed a discussion of low and high budget, independently of active learning, will put the present work in greater context. We will modify our related work section accordingly, and cite the appropriate references.
>
> Specific comments raised by the reviewer:
>
> **Clarity**:
>
> To improve the clarity of our paper, we will add pseudo-code for the second part of the algorithm to the appendix, and will reference it in the main text.
>
> **Scalability**:
>
> Empirically, our proposed method -- SelectAL -- demonstrates robustness and scalability to real-world applications. The total cost of SelectAL involves training a neural network once on two variants of the labeled set, denoted in the text by $\mathbb{L}$. While training two networks incurs this overhead, it is comparable to the training cost of the model that would use the labeled data afterward. Hearing the reviewer's concern and in order to further reduce the method's cost, in our future work we will examine the option to perform the test in Alg. 1 using a significantly smaller model. This discussion will be added to the summary and discussion section.
>
> It is worth clarifying that as currently presented, SelectAL already chooses a single low-budget and a single high-budget strategy to represent a larger family of active learning algorithms. In this way, we reduce the complexity of the method, as users are not required to separately examine each AL strategy they wish to consider. This already significantly contributes to reducing the method's computational cost. This heuristics was evaluated empirically, and the results reported in the paper show its effectiveness. After SelectAL returns its trinary decision, a user may decide to select a few uncertainty strategies, or a mixture of such strategies, if the high-budget regime is identified.
>
> **Combining low and high-budget strategies**:
>
> In a single round within the active learning framework, our theoretical analysis in Section 2.3 indicates that when the number of added examples is relatively small as compared to the total number of examples in the dataset, there is no clear advantage to mixing strategies, and specifically to combining low and high-budget strategies. Empirically, we validated this finding through a grid search, which showed that while combining several strategies does offer a minor improvement, it does not yield a significant boost in learning beyond any single active learning strategy. As a result, and since the use of multiple strategies adds complexity to the suggested approach, we propose to use of a single strategy.
>
> We note that the scenario changes in the context of multi-round active learning. With SelectAL's capability to switch strategies in each round, we can continually build the final labeled set, incorporating examples from different strategies along the way. As shown in Fig. 5, this dynamic approach leads to more substantial performance improvements as compared to any single uncertainty-based strategy. Thus there is a trade-off between the number of rounds and performance, where improved performance can be obtained with increased complexity by way of additional rounds, and when fewer examples are chosen in each round. We will add a discussion to this effect in the camera-ready revision.

---

> > ### Comment · Reviewer_uJGw · 2023-08-14
> > **Thank you for the clarifications.**
> >
> > Thank you for the clarifications.

---

### Author Rebuttal · Authors · 2023-08-07

We thank all the reviewers for the significant time and effort they made in reviewing our work.

In the attached PDF, we added an experiment suggested by Reviewer GcEN, in which we use other active learning strategies, namely ProbCover [1] and CoreSet [2], as the decision rules for SelectAL. Specifically, this experiment can be viewed as an extension of Figure 6 in the original manuscript, showing that SelectAL will make similar choices for a variety of active learning strategies. We believe that this addition enhances the comprehensiveness of our findings and strengthens the overall contribution of our work.

---

[1] Yehuda, Ofer, et al. "Active learning through a covering lens." Advances in Neural Information Processing Systems 35 (2022): 22354-22367.

[2] Sener, Ozan, and Silvio Savarese. "Active Learning for Convolutional Neural Networks: A Core-Set Approach." International Conference on Learning Representations. 2018.

---

### Comment · Area_Chair_cXoz · 2023-08-13
**Author-Reviewer Discussion**

Thanks to the authors for submitting a detailed rebuttal. Can the reviewers please read the author response and let them know if they have any further comments / need any additional clarifications?

Best regards,
 - Your AC.

---

### Decision · Program_Chairs · 2023-09-21

**Decision:**

Accept (poster)

**Comment:**

This paper was reviewed by four experts in the field and received one Weak Accept and three Borderline Accept as the ratings. The reviewers have agreed that the paper addresses an important problem in active learning research experimental results demonstrate the effectiveness of the proposed method.

The reviewers have mentioned that the theoretical analysis and the presented algorithm seem to be loosely connected, which has been addressed by the authors in the rebuttal. Concerns were also raised about the inconsistencies in the experimental setup and budget settings, which were also addressed convincingly by the authors in the rebuttal.

The reviewers, in general, have a positive opinion about the paper and its contributions. Most of them have been satisfied with the authors’ rebuttal and have recommended acceptance. Based on the reviewers’ feedback, the decision is to recommend the paper for acceptance to NeurIPS 2023. The reviewers have provided some valuable suggestions, such as including a pseudo-code of the proposed method in the Appendix, clearly explaining the connection between the theoretical analysis and the algorithm proposed and adding a few more reference papers in the Related Work section. The authors are encouraged to address these in the final version of their paper. We congratulate the authors on the acceptance of their paper!